# Optimizing Molecular Minimal Residual Disease Analysis in Adult Acute Lymphoblastic Leukemia

**DOI:** 10.3390/cancers15020374

**Published:** 2023-01-06

**Authors:** Irene Della Starza, Lucia Anna De Novi, Loredana Elia, Vittorio Bellomarino, Marco Beldinanzi, Roberta Soscia, Deborah Cardinali, Sabina Chiaretti, Anna Guarini, Robin Foà

**Affiliations:** 1Hematology, Department of Translational and Precision Medicine, “Sapienza” University, Via Benevento 6, 00161 Rome, Italy; 2GIMEMA Foundation, 00182 Rome, Italy

**Keywords:** acute lymphoblastic leukemia (ALL), minimal/measurable residual disease (MRD), RQ-PCR, digital droplet PCR (ddPCR), next-generation sequencing (NGS), novel molecular markers, cell-free DNA (cfDNA)

## Abstract

**Simple Summary:**

Minimal/measurable residual disease (MRD) monitoring is a powerful and independent predictor of outcomes in both children and adult acute lymphoblastic leukemia (ALL). MRD monitoring enables patients’ stratification into different risk-adapted treatment arms; it guides treatment decisions in clinical practice, including stem cell transplantation, and represents an early marker of impending relapse. Real-time quantitative PCR is the most widely used molecular method for MRD assessment, but there are some limitations that new approaches may overcome. In this review, we discuss the most recent technological advances in MRD monitoring that are allowing to increase the number of evaluable patients and the levels of quantification and also have the potential to study different disease compartments.

**Abstract:**

Minimal/measurable residual disease (MRD) evaluation has resulted in a fundamental instrument to guide patient management in acute lymphoblastic leukemia (ALL). From a methodological standpoint, MRD is defined as any approach aimed at detecting and possibly quantifying residual neoplastic cells beyond the sensitivity level of cytomorphology. The molecular methods to study MRD in ALL are polymerase chain reaction (PCR) amplification-based approaches and are the most standardized techniques. However, there are some limitations, and emerging technologies, such as digital droplet PCR (ddPCR) and next-generation sequencing (NGS), seem to have advantages that could improve MRD analysis in ALL patients. Furthermore, other blood components, namely cell-free DNA (cfDNA), appear promising and are also being investigated for their potential role in monitoring tumor burden and response to treatment in hematologic malignancies. Based on the review of the literature and on our own data, we hereby discuss how emerging molecular technologies are helping to refine the molecular monitoring of MRD in ALL and may help to overcome some of the limitations of standard approaches, providing a benefit for the care of patients.

## 1. Introduction

Acute lymphoblastic leukemia (ALL) is a heterogeneous malignancy in terms of clinical manifestations, biological background, clinical course, and prognosis. It affects the early stages of hematopoiesis and more frequently the B cell lineage (75–80% in adults and 85–90% in pediatric patients) than the T cell lineage (20–25% in adults and 10–15% in pediatric patients). The acquisition of a series of genetic aberrations leads to impaired maturation, with an arrest in the differentiation process and an abnormal proliferation, generating a progeny of leukemic lymphoid blasts [1].

ALL is the most frequent neoplasm in childhood and also affects adults of all ages: between 2 and 5 years and above the age of 50 [2,3], with 60% of cases discovered in individuals below 20 years of age [4].

Despite the progressively improved 5-year survival rate [5,6,7,8,9,10], about 20% of children with ALL still relapse [11,12], and in adults, the risk of relapse affects up to 50% of patients [13,14,15].

In the modern era, the management of Philadelphia chromosome-negative (Ph-) B-ALL and T-ALL is based on multidrug chemotherapy regimens in order to obtain complete hematologic remission (CR) and a status of minimal residual disease (MRD) negativity with or without an allogeneic stem cell transplantation (SCT), plus an effective central nervous system prophylaxis.

The management of Ph+ ALL has also markedly changed over the years thanks to the advent of tyrosine kinase inhibitors (TKIs), with treatment programs that involve the use of a TKI (plus glucocorticoids and central nervous system prophylaxis), with or without deintensified chemotherapy in the induction phase, followed by monoclonal antibodies or multiagent chemotherapy with or without allogeneic transplantation [16].

Over the years, cooperative study groups, both for pediatric and adult ALL patients, have defined in their treatment protocols specific informative time points.

The concept of MRD refers to the proportion of remaining cancer cells among otherwise normal bone marrow or, more rarely, among circulating blood cells after any given treatment. In ALL, MRD monitoring has proven to be an independent prognostic factor and an important instrument for therapeutic decisions.

Several studies [17,18,19,20] have evaluated MRD levels by using either flow cytometry [17,18] or a molecular approach [19,20] at multiple time points during treatment, showing that MRD measurements in the first 3 months of treatment are the most informative for risk group assignment in childhood ALL, and MRD monitoring in the post-remission phase is more widely used in adult ALL.

Thus, MRD cutoff values were established, enabling patients’ stratification into different risk-adapted treatment arms. MRD is also utilized to monitor disease burden in the setting of SCT and represents an early marker of impending relapse at any time point during the course of the disease.

MRD assays are required to be highly sensitive (≥10−4) and to have broad applicability, accuracy, and reliability [21,22]. Furthermore, for molecular MRD analysis, its genetic target should be representative of the disease, expressed by all leukemic clones, stable over time (i.e., present at both diagnosis and relapse) to reflect the kinetics of the malignant clone during treatment, and sensitive [18]. Currently, the gold standard for molecular MRD analysis is polymerase chain reaction (PCR) amplification-based methods that use leukemia-specific (fusion gene transcripts) or patient-specific (immunoglobulin/T cell receptor (IG/TR) gene rearrangements) molecular markers [23,24,25,26,27,28].

These methods have been extensively standardized within the EuroMRD Consortium (www.euromrd.org, accessed on 28 December 2022), which established guidelines for the analysis and interpretation of real-time quantitative PCR (RQ-PCR) data in order to favor a homogeneous application of MRD studies within different treatment protocols for both pediatric and adult ALL patients [24,29].

In the present review, we will focus on how emerging molecular technologies may refine the molecular monitoring of MRD in ALL and overcome some of the limitations of standard approaches, such as: (1) difficulty in identifying a molecular marker and/or a sensitive patient-specific primer at the onset of disease in about 5–10%, (2) clonal evolution of IG/TR rearrangement patterns during the course of the disease and at relapse, (3) the need for diagnostic DNA for each MRD experiment, and finally, (4) RQ-PCR is not able to define precisely the amount of residual disease in those cases in which the disease burden is very low.

## 2. IG/TR Gene Rearrangements and Fusion Gene Transcripts

### 2.1. IG/TR Gene Rearrangements

Clonal rearrangements of IG/TR genes can be detected in >95% of ALL patients and are the most frequently used markers for MRD analysis [30]. IG/TR gene rearrangements are physiological events and are not directly linked to the pathogenesis of leukemia. In the case of the neoplastic transformation of a single lymphoid cell, all leukemic cells are supposed to contain the same rearranged clonal IG and/or TR gene(s), which can be exploited to detect a low number of ALL cells among a large number of normal lymphoid cells expressing gene rearrangements with different sequences.

To identify these molecular markers at diagnosis, genomic DNA derived from leukemic cells undergoes PCR amplification, and clonal PCR fragments [31,32] are sequenced by standard Sanger sequencing to define the junctional regions and to obtain complementary allele-specific oligonucleotide (ASO) primers for MRD monitoring, mostly performed by RQ-PCR. Amplification conditions and sensitivity testing for each ASO primer are established by a standard curve built on serial dilutions of diagnostic DNA in a DNA pool from healthy donors. The sensitive range reached by this method is about one leukemic cell in 100,000 (10^−5^) normal lymphoid cells. The standard curve built on the diagnostic material is used to quantify MRD in bone marrow (BM) or peripheral blood (PB) samples collected during and after treatment [33].

Notably, for biological (due to cell maturity stage) and technical reasons, it is not possible to monitor MRD by RQ-PCR in all patients, with a rate of failure of about 5–10%. Moreover, IG/TR gene rearrangements may change during therapy, potentially leading to false negative MRD results [34]. These changes may be due either to a clonal evolution or to ongoing secondary rearrangement processes [35].

Furthermore, the use of RQ-PCR can be limited by the availability of sufficient diagnostic material because the method is based on a comparison with a standard curve based on neoplastic DNA collected at the onset of the disease; this can limit the possibility of monitoring patients over a prolonged time period. Furthermore, a non-negligible fraction of patients with very low MRD levels is classified as positive not-quantifiable (PNQ) since the MRD resulted positive but is detected outside the quantitative range when compared with the standard curve of the RQ-PCR assay.

A better discrimination of these cases, which currently represents a challenge in clinical practice, is needed, particularly when MRD data guide therapeutic decisions.

### 2.2. Fusion Gene Transcripts

Chromosomal translocations are detected in about 40% of ALL patients. These genetic aberrations are ideal targets for MRD evaluation because they are leukemic cell-specific and extremely stable during the course of the disease [36,37]. The most frequent lesion in adults (25–30% of cases, with a further increase in the elderly population of up to 50% [38], and only 2–5% in pediatric age) is represented by the translocation t(9; 22) (q34; p11), i.e., the Ph chromosome, which determines the *BCR-ABL1* rearrangement. The cryptic translocation t(12; 21) (p13; q22), responsible for the chimeric transcript *ETV6/RUNX1*, is very rare in adults (1% of cases), while it is observed in approximately 25–30% of childhood ALL, which is associated with a good prognosis.

Rearrangements of the *KMT2A* gene (alias *MLL*), located on chromosome 11q23, affect acute leukemias biologically unique for gene expression profiling. A number of genetic partners are currently known for the *KMT2A* gene. Among these, the *AFF1* gene, involved in the balanced translocation t(4; 11) (q21; q23), is present in 2–5% of cases regardless of age. Children less than 1 year (infants) carry a rearrangement involving the KMT2A gene in about 80% of cases.

In T-ALL, genetic lesions generally involve the long arm of chromosomes 14q11 (where the TRA and TRD genes are located) and 7q34 (where the TRB gene), resulting in the juxtaposition of the TR loci with different transcription factors.

The alteration that most frequently characterizes the *TAL1* gene is the deletion that juxtaposes the SIL gene (*SIL/TAL1*) and is detected in about 20% of cases [39]. Furthermore, other alterations involve ABL1; the more frequent rearrangements identified are NUP214, EML1, and ETV6.

Given the prognostic value of these chromosomal abnormalities, all cases at the onset of disease should be investigated [40] in order to monitor each patient by using a specific target.

Nevertheless, despite the great variability in the number of RNA transcripts produced, not all fusion genes are suitable for MRD evaluation [37]. In cases lacking a suitable MRD marker, alternative targets should be examined.

Given that the splicing process produces in all patients the same fusion transcript or a few splicing variants, the better material to detect these abnormalities is RNA. In this way, a small number of RQ-PCR assays is enough [41].

MRD assessment using fusion genes as a molecular marker is performed on a standard curve built on serial dilutions of a cell line or plasmid DNA (i.e., *BCR-ABL1*+).

The sensitive range reached is about one leukemic cell in 100,000 (10^−5^) normal lymphoid cells [37].

Despite its rapid and easy application, the accuracy of this analysis is altered by the variability in RNA transcripts, as already mentioned.

## 3. Novel Molecular Techniques for MRD Monitoring

### 3.1. Digital Droplet PCR (ddPCR)

ddPCR [42] appears to be a feasible and attractive alternative method for MRD assessment. The technique is based on the principle of partitioning the sample into several nano-PCR containing single, few, or no target sequences. PCR partitions are read as negative or positive by thresholds based on their fluorescence amplitude, and then the target sequence concentration is calculated by applying Poisson’s statistics [43,44] that link the theoretical depth of analysis to the number of compartments generated [45]. ddPCR [46,47] allows the quantitation of nucleic acid targets without the need for calibration curves [48] and seems to have a more precise quantification than the standard method because, in each droplet, the ratio between the target DNA to reagents is higher and each PCR is amplified and read individually so that any changes in fluorescence signal are detected [49] (Figure 1). Several ddPCR systems have been developed [50,51,52,53,54] thanks to their easy usage and applicability, adaptability, and saving of time.

At first, ddPCR was applied in the setting of molecular oncology [48,55] and prenatal diagnosis [56,57].

Currently, many reports on hematologic malignancies are also available [58,59,60,61,62,63,64]. The first studies established analytical parameters to investigate the applicability of ddPCR compared with RQ-PCR, and all reported a good concordance between the two techniques [58,59,60,61,62,63,64], underlining the sensitivity, accuracy, and reproducibility of ddPCR.

Regarding MRD evaluation, the discordances observed with RQ-PCR fell mostly at very low levels of disease. In this setting, the robustness of ddPCR to quantify RQ-PCR-PNQ samples and to identify false positive cases has been reported in several hematologic malignancies.

The Fondazione Italiana Linfomi (FIL) MRD Network has shown that non-Hodgkin’s lymphoma samples are defined as borderline because of alternating positive and negative results regardless of the method used (nested-PCR or RQ-PCR), and the type of rearrangement did not have interlaboratory discordance by ddPCR analysis [65]. Drandi et al. [66] compared ddPCR with RQ-PCR in mantle cell lymphoma MRD samples, showing very good concordance with both methods, particularly for samples with at least a 0.01% positivity. However, ddPCR presented a more robust quantification for low positive samples.

In ALL, few studies have explored the utility of ddPCR application for MRD assessment. In 2016, our group compared the two quantification techniques in 50 ALL adolescent patients, reporting a sensitivity and accuracy for ddPCR superimposable to that of RQ-PCR [63]. In 2019, we sought to analyze only adult ALL samples with RQ-PCR MRD levels ≤10^−4^ by both ddPCR/NGS [67]. The comparison showed a concordance rate of 57% and 52% for RQ-PCR/ddPCR and RQ-PCR/NGS, respectively, for samples concordantly positive or negative, while ddPCR and NGS also allowed to identify an MRD signal in very low positive samples, with a concordant MRD result among ddPCR and NGS in 87% of samples. The combined use of ddPCR and NGS significantly reduced the cohort of PNQ samples compared with RQ-PCR analysis and helped to predict three relapses in patients who resulted PNQ/NEG by RQ-PCR.

Recently, we also demonstrated that by increasing the number of patients and samples from a single trial, ddPCR was able to improve the rate of quantification in critically low positive samples [68]; moreover, the rate of concordance between ddPCR and NGS compared with RQ-PCR was also improved (92% vs. 87%).

These data demonstrated that in ALL, ddPCR is an extremely accurate tool when the RQ-PCR quantitative range is inferior to 10^−4^ thanks to its greater amplification efficiency and reproducibility. Moreover, the combined use of new technologies could better discriminate low positive samples that the standard method fails to detect or quantify the disease level [68].

Along the same line, Schwinghammer et al. [69] confirmed the high reproducibility and accuracy of ddPCR, particularly in very low positivity ranges and in quantitative ranges of higher positivity levels.

As for Ph+ ALL, three studies employed ddPCR for MRD monitoring [61,70,71]. In 2014, Iacobucci et al. [70] analyzed 60 ALL Ph+ samples, showing that ddPCR had a sensitivity to detect the disease comparable to that of the conventional approach.

A second study conducted by Coccaro and colleagues [61] proved that the ddPCR assay for p190 was able to quantify low disease levels by loading a high quantity of cDNA in different wells and combining the counts from multiple replicates. The higher ddPCR sensitivity was advantageous, translating in a timely manner for patient cure and predicting molecular relapse [61].

Finally, Ansuinelli and colleagues [71] tested 10 Ph+ ALL patients enrolled in the GIMEMA LAL 2116 trial [72], showing the ability of ddPCR to reduce the rate of PNQ samples and increase the proportion of quantifiable ones (46% of cases, *p* < 0.0001) significantly. Moreover, the authors highlighted as five cases that were RQ-PCR-negative and ddPCR-positive during the follow-up, 4/5 experienced a relapse [71].

As a step forward, there is growing interest in Ph+ ALL to monitor MRD using a “double-hit” strategy by using IG/TR gene rearrangements and the *BCR-ABL1* fusion transcript as molecular markers [73,74,75]. In this context, we are testing both targets by ddPCR in adult Ph+ ALL with the aim of defining the concordance rate during MRD monitoring in terms of sensitivity and specificity (ongoing) and once follow-up data will be mature, to finally dissect which marker is more reliable.

In a study conducted on a pediatric cohort of ALL patients, it was shown that ddPCR MRD quantification at a particular time point (i.e., day +78) was associated with relapse and, in line with this, ddPCR MRD negative or PNQ results at the same time point were positively associated with a better outcome, highlighting the possible clinical significance of the method [76].

Currently, there are no guidelines for ddPCR MRD application. However, the EuroMRD Consortium is actively working to rapidly achieve this goal. SOPs (standard operating procedures) are published as a guide for digital analysis in lymphoid malignancies [77,78,79].

### 3.2. Next-Generation Sequencing (NGS)

NGS technologies have, in general, represented a step forward in genomic analysis.

Their application can be subdivided into two main purposes: the first one is to identify novel molecular markers that could help toward improvement in disease biology comprehension, and a clinical practical phase for detecting alterations that could contribute to better disease management by improving risk stratification and disease monitoring for the benefit of patients.

The NGS approach is a very complex analytic method capable of providing a large amount of data. Over the years, several sequencing techniques have been developed with different applications: whole genome sequencing (WGS), transcriptome sequencing (RNA-seq), whole exome sequencing (WES), and targeted gene sequencing.

Beyond the abovementioned goals, a great effort is ongoing to define whether the genomic lesions identified by NGS technologies can be used for MRD monitoring.

NGS of IG/TR gene rearrangements (IG/TR-NGS) [21] allows for the quick screening of all patients with lymphoid malignancies without the need to use patient-specific assays for MRD analysis. The IG/TR-NGS clonality assay is based on a multiplex PCR to amplify the target regions and by subsequent ligation of adapters for sequencing. After purification of the PCR products, the library preparation is performed, followed by sequencing on the platform (e.g., Ion Torrent or Illumina) [80]. The Ion Torrent platform [81] makes use of the electrochemical detection of hydrogen ions that are released during DNA synthesis [82]. In contrast, Illumina employs fluorescently labeled nucleotides that are incorporated during complementary DNA strand synthesis. Depending on the type of Illumina sequencer, this can be a 2-channel (e.g., MiniSeq, NextSeq, Nova-Seq) or a 4-channel chemistry (e.g., MiSeq, HiSeq). Ion Torrent and Illumina sequencing technologies require specific adapters for sequencing and barcodes for sample identification: for Ion Torrent sequencing, the adapters and barcodes are ligated to the amplicon, while for Illumina sequencing platforms, the sequencing adapters need to be incorporated in the amplicon primers [82,83]. These methods allow to detect and sequence any possible IG/TR gene rearrangements with a sensitivity that ranges from 10^−4^ to 10^−6^ [21,84,85] (Figure 2); to reach higher sensitive levels, a higher amount of DNA is needed [86,87]. Thanks to its great sensitivity, this approach can identify more gene rearrangements compared with the standard method. Oligoclonality is a well-known phenomenon in ALL that hampers conventional IG/TR MRD [88] assessment, but this can be better identified by NGS. Moreover, multiple IG/TR gene rearrangements in ALL result from both continuing rearrangement processes and from secondary rearrangements [89,90,91,92].

To perform an accurate MRD analysis by NGS, it is mandatory to define which rearrangements, among those identified at diagnosis, should be selected [93] because diagnostic clonal sequences can be detected in follow-up samples.

Moreover, it must be reminded that by NGS, it is possible to detect early clonal evolution, which is a probable event in ALL relapsed cases [94,95]; however, at relapse, the most representative clone is the diagnostic one in the majority of cases.

A comparative analysis of NGS and RQ-PCR using IG/TR gene rearrangements is reported by some authors, showing a good correlation of the two methods but with higher sensitivity and specificity of NGS for MRD quantification [85,86,96]. Importantly, NGS analysis does not define cases as PNQ because, unlike standard methods, its quantitative range is always superimposable to the sensitive one [86]. Finally, the predictive value after induction therapy and after allogenic SCT of NGS evaluation has also been reported [85,96].

At present, however, there are no international guidelines for NGS MRD analysis.

In the last years, the EuroClonality-NGS working group has published several papers on the development and standardization of IG/TR-NGS assays, focusing on marker identification protocols for subsequent MRD analysis, with a multicenter validation across many expert European laboratories [97,98,99,100].

Lastly, the first NGS assay (ClonoSEQ) for MRD evaluation in ALL or multiple myeloma patients using IG gene rearrangements has been approved by the FDA [101].

Finally, NGS LymphoTrack assays to study IG/TR gene rearrangements have been developed by Invivoscribe (Invivoscribe, San Diego, CA, USA). Different reports have demonstrated the clonality concordance between the LymphoTrack system and other molecular approaches and with disease monitoring, allowing to define the tool as sensitive and accurate for diagnostic testing and disease monitoring in B and T cell tumors [102,103,104].

Moreover, compared with the standard method, NGS assays show superior detection in cases with a lower tumor burden, allowing a higher resolution of clonal rearrangements partially hidden in a polyclonal background [105].

Thus overall, the higher sensitivity, specificity, and application of NGS for MRD monitoring compared with standard approaches make its implementation in the clinical practice one of the most important next goals to achieve. The availability of skilled bioinformatics, however, is required.

Some authors have compared ddPCR and NGS approaches in the setting of SCT [106,107], defining the methods as easy to perform and user-friendly in all molecular biology laboratories and highlighting a high correlation and concordance between the results [107].

In our experience, as already mentioned in the previous section, the combined use of both ddPCR and NGS significantly reduced the number of critical samples defined as PNQ by RQ-PCR, which is a challenge in clinical practice, showing great concordance with MRD quantification [66,67].

## 4. Novel Compartments for MRD Monitoring

MRD assessment is usually carried out on BM cells. In T-ALL, PB may be a reliable source for MRD monitoring since there appears not to be significant differences with the BM and could therefore be used as an alternative source. On the contrary, in B-lineage ALL, MRD levels tend to be 1 to 3 logs lower in PB compared with BM [108,109].

The study of cell-free DNA (cfDNA) and other PB components (known as “liquid biopsies”) is promising and has been investigated, especially in solid tumors [110]. A “liquid biopsy” can define the molecular profile of a tumor by analyzing PB using different methods: cell-free circulating nucleic acids (DNAs, mRNAs, micro-RNAs, or noncoding RNAs), circulating tumor cells (CTCs), or exosomes [111] (Figure 3).

**cfDNA,** composed of extracellular short double-stranded DNA fragments and detectable in almost all body fluids, including the blood, is involved in various physiologic and pathologic phenomena [112]. The majority of cfDNA found in the plasma originates from leukocytes, and only a small fraction is tumor-derived, defined as circulating tumor DNA (ctDNA). The concentration of ctDNA varies among patients and differs according to the type, localization, and stage of cancer.

MicroRNAs (miRNAs) are the most abundant RNA molecules in the blood. They present high stability and are very important in tumor growth and treatment resistance [113].

These cell-free circulating nucleic acids (cfDNAs and miRNAs) have been investigated in solid cancers and in non-Hodgkin’s lymphomas and are currently being investigated in ALL.

CTCs derive from an early step of tumor blood dissemination and are detected at low levels in most patients.

Studies of both animal models and cancer patients have demonstrated that CTCs have the potential to facilitate the early detection, prognosis, therapeutic target selection, and monitoring of response to treatment [114].

Exosomes are detectable in some types of cancer, carrying proteins and nucleic acids.

They are analyzed through their RNA content [115]. Compared with cfDNAs and miRNAs, CTCs and exosomes have found applications in other hematologic diseases, such as lymphomas, multiple myeloma, and chronic lymphocytic leukemia.

Recently, many studies also evaluated the potential use of cfDNA for risk stratification, monitoring tumor burden, and response to treatment in hematologic malignancies [116,117,118,119,120], with the caveat that the amount of cfDNA can be very low and variable across patients, and this can represent a challenge for MRD analysis. Its usage in ALL has been explored in some studies.

Schwarz and colleagues [121] conducted a study aimed at testing the feasibility of MRD measurements employing cfDNA from children with ALL. The authors concluded that despite low concentrations, cfDNA is a feasible tool to measure MRD kinetics during ALL induction therapy.

Likewise, Cheng et al. [122] evaluated the feasibility of MRD measurement by IG/TR gene rearrangement quantification on cfDNA, comparing it to flow cytometry analysis on BM. The study showed not only the feasibility of using cfDNA as a tool for treatment monitoring in leukemia patients but also demonstrated its prognostic significance.

Indeed, one of the most intriguing applications of cfDNA is represented by the evaluation of extramedullary relapses, in which leukemic clones are rarely sampled by standard BM aspirate/biopsy [123,124,125].

While liquid biopsy is an extremely attractive tool, it must be kept in mind that several preanalytic factors affect ctDNA amount and quality, thus hampering, at present, usage for molecular analysis [126,127] and highlighting the need to develop a standardized protocol to improve the impact of preanalytical step variability during plasma sample processing and, at the technological level, to allow accurate, robust, and reproducible results of ctDNA genotyping and quantification.

## 5. Conclusions

MRD is a powerful and independent predictor of outcomes in both children and adult ALL. MRD can provide different information according to the timing in which it is performed: very early during treatment, after induction/consolidation, or before and after SCT. Currently, RQ-PCR of IG/TR gene rearrangements and gene fusions is the most broadly applied and consolidated method for molecular MRD monitoring. Despite its broad applicability, the method, however, presents intrinsic limitations that must necessarily be overcome. ddPCR and NGS appear to be feasible and attractive alternative approaches for MRD assessment, more sensitive and accurate than the standard method that can better discriminate critical samples by RQ-PCR analysis, particularly in certain subsets, such as early T-ALL and pro-B ALL, often lacking a sensitive molecular marker. Given their intrinsic technical features, the combined use of both approaches could provide an accurate MRD analysis for virtually all patients. A major effort is ongoing at the international level to standardize and better understand how these new techniques could be incorporated into clinical trials.

During the last years, great interest has focused on the possibility of investigating ctDNA as a noninvasive, real-time, and sensitive tool. ctDNA may, in fact, provide a more comprehensive molecular overview of the genetic heterogeneity of a given tumor at presentation and may potentially enable sequential blood testing during the course of the disease. However, to prove the feasibility and reproducibility of ctDNA for the monitoring of MRD in patients with hematologic malignancies, some preanalytical biases that affect analysis need to be overcome.

MRD analyses are becoming progressively more refined and represent the main tool to design treatment protocols in ALL, always aimed at a more profound eradication of the disease and at curing more patients.

A word of concern is required, as the management of ALL patients of all ages is always becoming more laboratory-driven, from the diagnostic and prognostic work-up to the close monitoring of MRD during the different phases of the disease, leading to progressively more personalized treatment strategies. How doable is this worldwide? This aspect cannot be ignored, as this complex approach is paramount in order to offer optimal management to ALL patients and to increase the likelihood of a cure.

## Figures and Tables

**Figure 1 cancers-15-00374-f001:**
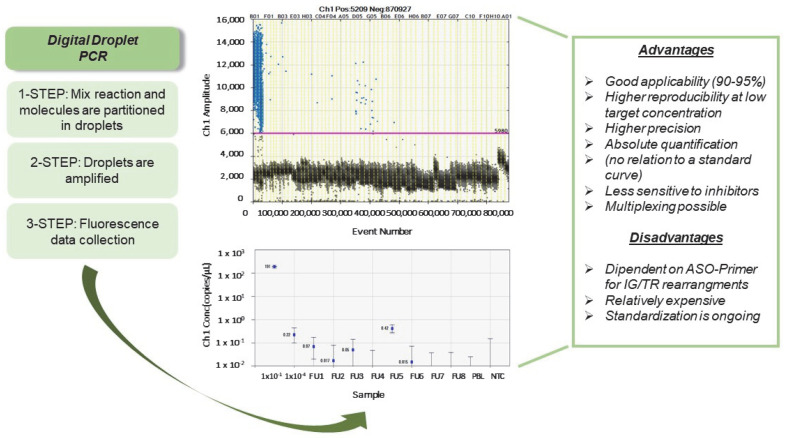
ddPCR technology. This figure reports an analytical diagram for a ddPCR experiment. The mix reaction and the DNA samples are partitioned into 20,000 droplets (**1-step**), and then both are amplified in a thermal cycler (**2-step**). Finally, each droplet is individually analyzed by a droplet reader (**3-step**). Each droplet is plotted on the graph of fluorescence intensity, and the concentration is calculated on the fraction of droplets that does not contain any target DNA using software. The positive droplets are fitted to a Poisson algorithm to determine absolute copy number expressed as copies per 1 μL.

**Figure 2 cancers-15-00374-f002:**
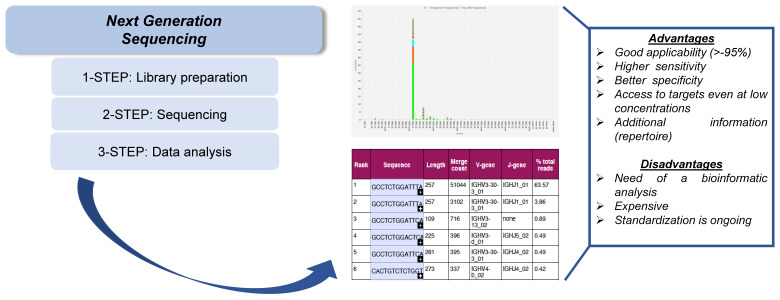
NGS analysis. A library is prepared by fragmentation and conjugation with adaptive sequences, composed with few nucleotides (**1-step**), and subsequently, it is amplified and sequenced, with the production of so-called ≪reads≫ (**2-step**). Data analysis is performed through the use of bioinformatic tools, matching the results to a reference genome (**3-step**).

**Figure 3 cancers-15-00374-f003:**
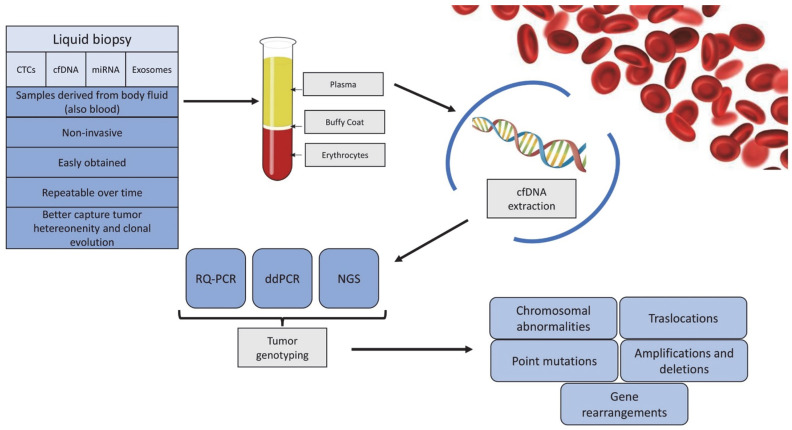
cfDNA as novel compartment for molecular MRD monitoring in ALL.

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
