# Peer review of "Optimizing Molecular Minimal Residual Disease Analysis in Adult Acute Lymphoblastic Leukemia"

_cancers, 2023, doi:10.3390/cancers15020374_

Round 1
Reviewer 1 Report
This review is well written and represent a valuable summary on the changing field of molecular MRD in the setting of adult acute lymphoblastic leukemia. I have minor observations concerning how the manuscript might be improved:
1. Could the authors try to delineate what is still necessary to include these technological advances in innovative MRD oriented clinical trials?
2. Do the authors believe that ddPCR or NGS may become an alternative or should be used in combination to RQ-PCR on a routine basis?
3. How do the authors expect the MRD landscape changing in the coming years? For example: will we need to learn new decisional time points or to modulate treatment intensity according to the level of MRD positivity?
4. Will there be specific setting of ALL patients for whom new modalities of MRD monitoring might be preferred, such as Ph positive or Ph-like ALL, or T-ALL?
5. There are some typos or unclear sentences: a) page lines 128- 131 The sentence is not well constructed and should be amended; b) page 5 lines 230-232 the sentence should be revised; page 9 lines 379-381 “…sampled by sampled by…” Please correct
Reviewer 2 Report
The review by Della Starza, et al summarizes the use of molecular methods to measure MRD in ALL. It is well organized and generally well written, although it could use some proof reading for English. It could be a good resource for readers. A few suggestions for improving it are provided below.
1. In the introduction around line 68 and forward, it will be very important to indicate which MRD method was employed for the references cited. Some if not all used a flow-cytometry based MRD assay. Many risk assignment and prognosis algorithms in use rely on flow cytometry rather than molecular detection. This is a very important point that should be clarified where relevant.
2. The last sentence of the introduction, line 90, refers to limitations of current technologies. Perhaps the authors could expand on such limitations in the introduction.
3. Line 168 – what is meant by “nano-PCR”? Do the authors mean to refer to droplets here? Also please clarify what is fluorescent in the ddPCR method.
4. The discussion of the two commercial NGS assays at the bottom of page 7 is rather brief. Since readers might want to use these available assays, it would be helpful if the authors could provide some context for them. For instance, what is the approved indication for the assay? In other words, how is it supposed to be used, for which patients, etc?
5. Overall, the discussion of these very sensitive molecular methods would be improved by an expanded review of the clinical prognostic relevance, or lack of relevance, for the very very low levels of MRD that can be detected. A more thorough discussion of the limitations of these technologies, such as cost, turnaround time, and other factors should also be included.
Minor errors:
1. Line 157 there is a typo at the end of the sentence: “nee.”
2. Line 230 – there is a sentence fragment.
